# Comparison of the Parameters of the Exergoeconomic Environmental Analysis of Two Combined Cycles of Three Pressure Levels with and without Postcombustion

**DOI:** 10.3390/e24050636

**Published:** 2022-04-30

**Authors:** Edgar Vicente Torres González, Sergio Castro Hernández, Helen Denise Lugo Méndez, Fernando Gabriel Arroyo Cabañas, Javier Valencia López, Raúl Lugo Leyte

**Affiliations:** 1Programa de Energía, Universidad Autónoma de la Ciudad de México–San Lorenzo Tezonco, Prol. San Isidro 151, San Lorenzo Tezonco, Iztapalapa, Ciudad de México 09790, Mexico; edgar.vicente.torres.gonzalez@uacm.edu.mx (E.V.T.G.); fernando.arroyo@uacm.edu.mx (F.G.A.C.); 2Departamento de Ingeniería de Procesos e Hidráulica, Universidad Autónoma Metropolitana–Iztapalapa, Av. Ferrocarril San Rafael Atlixco 186, 1 A Sección, Iztapalapa, Ciudad de México 09340, Mexico; sch@xanum.uam.mx (S.C.H.); lulr@xanum.uam.mx (R.L.L.); 3Departamento de Procesos y Tecnología, Universidad Autónoma Metropolitana–Cuajimalpa, Av. Vasco de Quiroga 4871, Santa Fé, Cuajimalpa, Ciudad de México 05348, Mexico; jvalencia@cua.uam.mx

**Keywords:** combined cycle, postcombustion, residue, environmental indicators, exergoeconomic operation costs

## Abstract

Nowadays, in Mexico, most of the installed electricity generation capacity corresponds to combined cycles, representing 37.1%. For this reason, it is important to maintain these cycles in good operating conditions, with the least environmental impacts. An exergoeconomic and environmental analysis is realized to compare the operation of the combined cycle, with and without postcombustion, with the comparison of exergoeconomic and environmental indicators. With the productive structure of the energy system, the process of formation of the final products and the residues are identified, and an allocation criterion is also used to impute the formation cost of residue to the productive components related to its formation. This criterion considers the irreversibilities generated in each productive component that participates in the formation of a residue. The compositions of pollutant gases emitted are obtained, and their environmental impact is determined. The unit exergoeconomic cost of the power output in the gas turbine is lower in the combined cycle with postcombustion, indicating greater efficiency in the process of obtaining this energy stream, and the environmental indicators of global warming, smog formation and acid rain formation are higher in the combined cycle with postcombustion, these differences being 5.22%, 5.53% and 5.30%, respectively.

## 1. Introduction

In Mexico, electricity generation with fossil fuels has been increasing by around 1.8% on an annual average. In 2018, 329,162 GWh were produced, of which 21.1% came from clean technologies and the remaining 78.9% from conventional thermoelectric plants, repowered and combined cycles. Power plants using fossil fuels emitted around 120 million tons of CO_2eq_ and it is estimated that in the year 2032 they will be increased to approximately 145 million tons of CO_2eq_ to meet the growing demand for electrical energy [1]. Combined cycles have become a core technology for converting the chemical energy of fossil fuels into shaft work used for electrical generators in the industries of electrical power. However, this conversion process produces a waste stream corresponding to the exhaust gases at high temperatures, which are released into the environment. The composition of these gases includes greenhouse gases such as carbon dioxide, and methane; contaminant gases, for example, carbon monoxide, nitrogen oxides, and unburned fuels. This stream is known as residue and its results are essential for evaluating the sustainability of the gas turbine operation, establishing recommendations to reduce the environmental impact of their operation, and recovering the energy content of the exhaust gases. In the search for actions that decrease the environmental impact of electricity generation, the thermodynamic, economic and environmental diagnosis of the power plants has been implemented, which is included in the exergoeconomic-environmental diagnosis.

The power generation costs are mainly influenced by energetic, economic, environmental, and toxicity factors. There are different types of methodologies used to estimate the energetic costs of power generation plants. Some of these methodologies are based on the application of the first and second law of thermodynamics and include the works of Lugo et al. [2,3,4], Kotas [5], and Dincer et al. [6], among others. Furthermore, there are also methodologies to estimate the power generation based on the exergoeconomic analysis of the gas turbine, which is the main subsystem of power plants, such as single gas turbine, combined cycle, and cogeneration plants, such as those pursued by Valero et al. [7,8], Tsatsaronis [9], Bejan et al. [10], and Torres et al. [11]. The environmental impact of power generation has also been studied by Hakan et al. [12], Goran et al. [13], and Dincer et al. [6] by accomplishing environmental, sustainability, and exergoenvironmental analyses applied to power generation plants. Exergoeconomic analysis is a technique to diagnose energy systems, through the allocation of costs to their energy streams, and arises from the combination of the second law of thermodynamics and the principles of economic engineering [14,15]. The exergoeconomic analysis of A. Valero et al. [14,16], also known as thermoeconomics, is based on the approach of the productive structure of an energy system, in which each component of the system has a productive objective and a required resource; and it starts with a resource external to the system, the product of a component is a resource of another component, and so on, until reaching the final product of the system, and then there is the process of cost formation, both of the final product and of residues [17,18,19]. In the exergoeconomic analysis, the formation of the products of an energy system is carried out through its productive components. However, these are always accompanied by the formation of flows of matter or energy that were not expected and even unwanted, known as waste. These wastes are exergetic losses that are discharged into the environment by means of dissipative components and can cause an impact on the environment, and their cost of formation is attributed to the product components that form them [20,21,22,23]. Research studies by S. Keshavarzian et al. and C. Torres et al. on different energy systems demonstrate the practical viability of the exergoeconomic approach when diagnosing these systems, and determining the impact on fuel consumption [15,16,24]. On the other hand, L. Meyer, G. Tsatsaronis et al. perform exergoenvironmental analysis, combining exergy analysis with the environmental impact of polluting gas emissions on the environment [25,26].

In this article, an exergoeconomic-environmental analysis is applied to a combined cycle with postcombustion, since these are the present and future technologies of the national energy panorama, to find a relationship between the use of resources, the environmental impact, and the exergoeconomic operating costs, and its behavior is also compared with that of a combined cycle without postcombustion, through energy, exergetic, exergoeconomic and environmental indicators. Furthermore, a criterion has been proposed to allocate the cost of residue formation to the productive components related to its formation. This criterion is based on the irreversibilities generated in each productive component that participates in the formation of a residue. The criterion is an extension of the entropy change criterion because, in addition to the entropy changes, it also includes the exergy flow rates associated with the transfer of heat through the limits of the energy system. In addition, the aim of this article is to present a methodology based on exergetic, exergoeconomic, environmental, and toxicity analysis to estimate the irreversibilities and exergetic efficiencies of each of the main components of a combined cycle with postcombustion, as well as to find the exergoeconomic operation costs and environmental and human toxicity indexes. This analysis is useful to evaluate each component, as well as the overall system; and to provide possible operation conditions enhancing the combined cycle performance and reducing its impact on the environment.

## 2. Materials and Methods

### 2.1. Description of the Combined Cycle

The combined cycle with postcombustion studied in this paper is integrated of a gas turbine, a heat recovery steam generator with three pressure levels and a steam cycle, as shown in Figure 1 [27]. The gas turbine consists of an air compressor, a combustion chamber and an expansion turbine. In the compressor, the suction air, *g*1, is compressed to a higher pressure and temperature, *g*2. Then, the air enters the combustion chamber, where combustion occurs by fuel injection. The combustion gases leave the combustion chamber and enter the turbine, *g*3; the hot gases are expanded in this turbine, where it generates a useful output power, and then these gases enter the HRSG, *g*4. In the HRSG, the combustion gases exchange heat, *g*4 to *g*15, with a steam cycle to generate output power in the steam turbine. In order to generate more steam, the temperature of the combustion gases is raised from *T**_g_*_5_ to *T**_g_*_6_ with a postcombustion, and in this way, the power output is increased. This postcombustion is carried out between the sections of the high-pressure superheater 1–intermediate pressure reheater 1, HPSH1 + IPRH1, and the high-pressure superheater 2–intermediate pressure reheater 2, HPSH2 + IPRH2. The pressure and temperature conditions of the steam cycle are equal to the CC conditions without postcombustion [27], and the pinch-point temperature differences are: Δ*T_ppHP_ = T_g_*_8_ − *T_satHP_*, Δ*T_ppIP_ = T_g_*_12_ − *T_satIP_* y Δ*T_ppLP_ = T_g_*_14_ − *T_satLP_*. Compared to a CC without postcombustion, this system generates more superheated steam in the HRSG and consequently more power output. 

#### Thermodynamics of the Combined Cycle

In this article, the combined cycle with postcombustion assumes air and combustion gases as perfect gases, the principles of conservation of mass and energy are applied, and changes in kinetic and potential energy are neglected. With environmental conditions, the volumetric composition of the fuel, the compressor pressure ratio, the gas turbine inlet temperature, the pressure drops in the heat exchangers, the steam cycle temperatures and pressures and the generated power by the gas turbine, the thermodynamic states of the system are determined and the energy and exergetic performance indicators are deduced, as well as the exergetic flow rates, in Appendix A the mathematical models of the GT are presented and in Appendix B, for the HRSG and SC.

From this information, the thermodynamic states of the system are determined and the energy and exergetic performance indicators are deduced, as well as the exergetic flows, in Appendix A the mathematical models of the GT are presented and in Appendix B, for the HRSG and the SC. Figure 2 and Figure 3 show the thermodynamic states of the plant in the exergy–enthalpy diagrams of the GT cycle with the HRSG (gas side) and of the steam cycle, respectively. This gas turbine operates on the thermodynamic cycle presented in the exergy–enthalpy diagram of Figure 2, in which air entering the compressor at state g1 is compressed to some higher pressure at state *g*2. Leaving the compressor, air enters the combustion chamber, where combustion occurs by fuel injection. The combustion gases leave the combustion chamber and enter the turbine, *g*3*,* where these gases are expanded to state *g*4 to generate the useful output power. Since the highest temperature and pressure reached in the gas turbine correspond to the turbine inlet state, *g*3, this state presents the maximum exergy. In the HRSG, the combustion gases exchange heat, *g*4 to *g*15, with a steam cycle. Furthermore, the postcombustion in HRSG increased the combustion gas exergy, *ε_g_*_6_
*− ε_g_*_5_. In the steam cycle, steam flows to a steam turbine to generate mechanical energy, which is used to drive an electrical generator. The reduced-energy steam flows out of the turbine and enters the condenser, where it is condensed to the condition of saturated liquid. A feedwater pump returns the condensed liquid to the heat recovery steam generator. The steam cycle operates in agreement with the exergy–enthalpy diagram depicted in Figure 3. Even if the streams of main and reheated steam, *v*1 and *v*4, have the same temperature, *v*1 presents the highest exergy content while the state *v*4 has the greatest energy content. The pressures associated with each of these states explain this fact since the pressure of *v*1 is greater than the pressure of *v*4. In addition, the exergy of the heat discarded in the condenser, *ε_v_*_6_
*− ε_v_*_7_, is low, even if its energy content, *h_v_*_6_
*− h_v_*_7_, is high because the condensation temperature is very close to the temperature of the surroundings (the dead state temperature). On the other hand, the exergy changes in the steam expansions, *ε_v_*_1_
*− ε_v_*_2_ and *ε_v_*_4_
*− ε_v_*_6_, are greater than their energy changes, *h_v_*_1_
*− h_v_*_2_ and *h_v_*_4_
*− h_v_*_6_, indicating that only a part of the steam exergy is used to generate work.

The combustion reaction of natural gas C*_n_*H*_m_* with air atmospheric as oxidant element is the atmospheric air. The mass fractions of combustion gases CO, NO*_x_*, CO_2_, and unburned (C*_n′_*H*_m′_*) are determined by the equations of Rizk and Mongia [28]. Then, the combustion reaction of hydrocarbons, C*_n_*H*_m_*, with atmospheric air as an oxidant element can be written as:(1)$$\begin{array}{ll} c_{n}H_{m}+ & X_{estVA}(1+\lambda )\left[X_{DA}\left(0.21O_{2}+0.79N_{2}\right)+X_{H_{2}O}H_{2}O\right]\\ & \to \theta_{1}{\mathrm{CO}}_{2}+\theta_{2}H_{2}O+\theta_{3}N_{2}+\theta_{4}O_{2}+\theta_{5}\mathrm{CO}+\theta_{6}c_{n}H_{m^{\prime }}+\theta_{7}{\mathrm{NO}}_{x} \end{array}$$

Figure 1 shows that supplementary firing takes place between HPSH1 + IPRH1 and HPSH2 + IPRH2 of the HRSG, allowing to raise the gas temperature from *T_g_*_5_ to *T_g_*_6._

The afterburning of natural gas is performed only with the exhaust gases because these are rich in oxygen due to the excess air supplied to the gas turbine, therefore, the fraction of oxygen contained in exhaust gases is utilized for postcombustion. The postcombustion reaction can be expressed as:(2)$$n_{c_{n}H_{m}}c_{n}H_{m}+X_{{\mathrm{CO}}_{2}}{\mathrm{CO}}_{2}+X_{H_{2}O}H_{2}O+X_{N_{2}}N_{2}+X_{O_{2}}O_{2}+X_{\mathrm{CO}}\mathrm{CO}+X_{c_{n}H_{m}}c_{n}H_{m}+X_{{\mathrm{NO}}_{x}}{\mathrm{NO}}_{x}\;\to \delta_{{\mathrm{CO}}_{2}}{\mathrm{CO}}_{2}+\delta_{H_{2}O}H_{2}O+\delta_{N_{2}}N_{2}+\delta_{O_{2}}O_{2}+\delta_{\mathrm{CO}}\mathrm{CO}+\delta_{c_{n^{\prime \prime }}H_{m^{\prime \prime }}}c_{n^{\prime \prime }}H_{m^{\prime \prime }}+\delta_{{\mathrm{NO}}_{x}}{\mathrm{NO}}_{x}$$

The molar fraction of each component of exhaust gases of the combined cycle with postcombustion is
(3)$$X_{i}=\frac{\delta_{i}}{\sum_{i}\delta_{i}}$$

The molar mass of the flue gas is
(4)$$MM_{cg}=\sum_{i}X_{i}\;MM_{i}$$

The mass fraction of each component of exhaust gases of the combined cycle with postcombustion, is given by:(5)$$f_{i}=\frac{X_{i}\;MM_{i}}{MM_{cg}}$$

The gas natural volumetric composition utilized in this study is 88% [CH_4_], 9% [C_2_H_6_] and 3% [C_3_H_8_] [29,30], whose reduced formula is C_1.15_H_4.3_, and its low heating value is 49,494.82 kJ/kg*_f_*. Table 1 gives the values of the volumetric composition and fraction of the exhaust gases from the combined cycle with postcombustion, where the polluting gases are CO_2_, CO, unburned (C*_n_*H*_m_*) and NO*_x_*. The molecular weight of the exhaust gases is 18.1 kg/kmol.

The performance indicators for the combined cycle with postcombustion derived from thermodynamic analysis are the thermal and exergetic efficiencies, which can be written in a general form as follows: (6)$$\eta_{thCC}=\frac{\;{\dot{W}}_{mGT}+{\dot{W}}_{mSC}}{{\dot{m}}_{a}c_{Pa}\;T_{g1}\left[(1+far)\;\frac{c_{Pcg}}{c_{Pa}}\;y-1-\frac{1}{\eta_{sic}}\left(\pi_{C}{}^{x_{a}}-1\right)\right]+{\dot{m}}_{f2}LHV}\;\mathrm{and}\;\eta_{exCC}=\frac{{\dot{W}}_{mGT}+{\dot{W}}_{mSC}}{\left({\dot{m}}_{f1}+{\dot{m}}_{f2}\right)LHV\left(1-\frac{T_{0}}{T_{af}}\right)}$$

### 2.2. Exergoeconomic Evaluation

Based on the exergy flow rates of the energy streams and the irreversibility flows of the system components, the productive structure of the system is proposed; where the external resource of the system is distributed in its components, and each of the components has its own product and resource; this product is distributed as a resource for other components, to obtain its respective product, and so on until the final product and residue of the system are reached. In addition, each component has an associated irreversibility, and as a consequence, the irreversibilities accumulate in the process of obtaining each stream in the system. In this way, the process of formation of the final product is identified and in parallel, the process of formation of the residue, and the product components are identified, which have a defined product, and the dissipative components, which are characterized by the absence of a productive purpose but its usefulness lies in the dissipation of residue into the environment and its presence is important due to the interaction they have with other components to obtain the final product of the system. To carry out the exergoeconomic analysis, based on Figure 1, the productive structure of the CC with postcombustion is proposed, which appears in Figure 4. This indicates that the external resources are air, the fuel supplied in the combustion chamber in postcombustion, and the water; the products are the powers output by the GT and SC; and that the residues are the exhaust gases and the heat dissipated in the condenser.

The exergoeconomics determines the cost of production of a system, considering the process of product formation. However, this process is always accompanied by residues, which are energy streams or materials that are not in balance with the environment, in other words, there are thermodynamic potentials; These residue streams are wasted resources. Currently, there is no established methodology to impute the costs of residue since its formation process impacts on the one hand the cost of production and on the other hand the cost of residue. In this work, it is being imputed by means of a proposed criterion of irreversibility, where it is assumed that the productive components contribute to the cost of residue formation, in the proportion of their irreversibility generated in relation to the total irreversibility of the system. This criterion is based on accounting for irreversibilities throughout the formation process of the residues. For the combined cycle, the formation of exhaust gases is accompanied by the generation of irreversibilities in the GT and HRSG processes, on the gas side, to produce work and heat. Whereas the heat dissipated in the condenser is the result of the irreversibilities in the steam generation processes in the HRSG and the power output in the productive components of the steam cycle. With the application of the allocation rules to the components of the combined cycle [26], the exergetic and exergoeconomic cost balances are made, considering a ratio of allocation to the product components for the formation of the residue costs as a function of their internal irreversibilities. Then, for the *k-*th component of the combined cycle, proposition 2 [23] of the allocation rules is applied as:(7)$$\Pi_{F_{k}}^{}+\mu_{k}^{{\dot{I}}_{prod}}\;\Pi_{g15}^{}+\beta_{k}^{{\dot{I}}_{prod}}\;\Pi_{{\dot{Q}}_{COND}}^{}=\Pi_{P_{k}}^{}$$
where the fractions of allocation of the cost of formation of the exhaust gas residues, $\mu_{k}^{{\dot{I}}_{prod}}$, and the heat dissipated in the condenser, $\beta_{k}^{{\dot{I}}_{prod}}$, are expressed as follows
(8)$$\mu_{k}^{{\dot{I}}_{prod}}=\left\{ \begin{array}{l} \frac{{\dot{I}}_{k}}{{\dot{I}}_{cg}^{prod}}\;;\;k\in GT\\ \frac{{\dot{F}}_{HRSG,\;k}-{\dot{E}}_{{\dot{Q}}_{COND}}}{{\dot{I}}_{cg}^{prod}}\;;\;k\in HRSG \end{array}\right.\;\mathrm{and}\;\beta_{k}^{{\dot{I}}_{prod}}=\left\{ \begin{array}{l} \frac{{\dot{E}}_{{\dot{Q}}_{COND}}-{\dot{P}}_{HRSG,\;k}}{{\dot{I}}_{v}^{prod}};\;k\in HRSG\\ \frac{{\dot{I}}_{k}}{{\dot{I}}_{v}^{prod}}\;;\;k\in SC_{prod} \end{array}\right.$$
and the irreversibility flows rate are:(9)$${\dot{I}}_{cg}^{prod}={\dot{I}}_{GT}+{\dot{I}}_{Poscombustion}+{\dot{F}}_{HRSG}-\sum_{k\in HRSG}{\dot{E}}_{{\dot{Q}}_{k}}\;\mathrm{and}\;{\dot{I}}_{v}^{prod}=\sum_{k\in HRSG}{\dot{E}}_{{\dot{Q}}_{k}}-\;{\dot{P}}_{HRSG}+{\dot{I}}_{SC}$$

**Figure 4 entropy-24-00636-f004:**
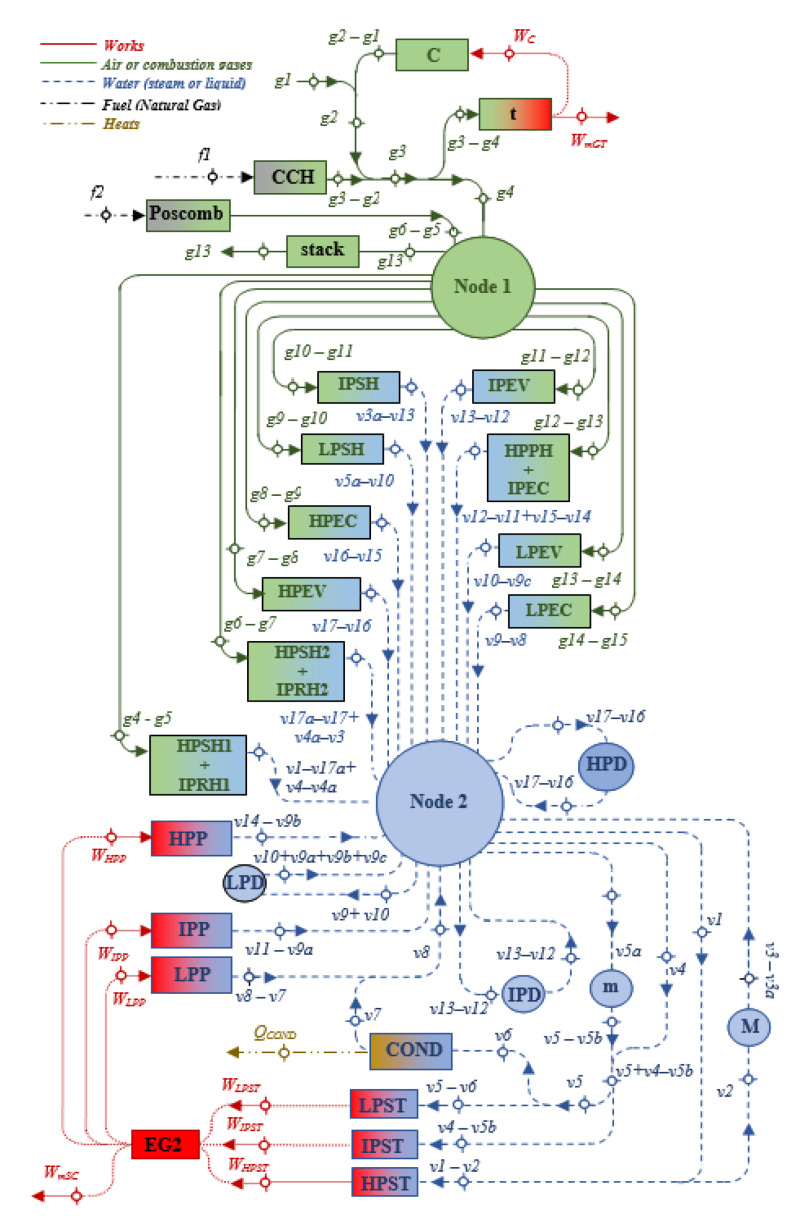
Productive structure of the combined cycle with postcombustion.

With Equation (8), and under the criterion of irreversibility, it is observed that the productive components of the steam cycle do not contribute to the process of formation of the exhaust gas residue, and for the process of formation of the residual heat dissipated in the condenser, the components of the gas turbine do not contribute. However, with Equation (9), this criterion considers the irreversibility of the gas and steam sides within the HRSG. To address this problem, the level of aggregation in the HRSG is reduced, in such a way that each of its components has two subsystems (one for gases and one for steam) that exchange heat. The coupling of the GT and the SC takes place in the HRSG. In this study, the components of the HRSG are treated as adiabatic heat exchangers, where the thermal energy of the combustion gases is used for the generation of steam. The components of the HRSG are considered as subsystems made up of two thermal energy deposits that exchange heat (${\dot{Q}}_{\;k}$), one corresponding to combustion gases and the other to steam. For exhaust gases, ${\dot{E}}_{{\dot{Q}}_{k}}$ is the product, and for steam, it is the resource.

The exergoeconomic costs, presented in Table 2, Table 3 and Table 4, are derived by carrying out exergoeconomic balances to each component and stream of the system, based on Figure 4, as well as by using the propositions of the exergoeconomic theory.

The exergoeconomic operating cost is the cost associated with the destruction of exergy in a process (irreversibility), this is the product of the unit exergoeconomic cost of the resource and the difference between the exergy flow rates of the resource and the product of each of the system components (irreversibility):(10)$$EOC_{k}=c_{F_{k}}\left({\dot{F}}_{k}-{\dot{P}}_{k}\right)\;\mathrm{where}\;c_{F_{k}}=\frac{\Pi_{F_{k}}^{}}{{\dot{F}}_{k}}$$

### 2.3. Environmental Indicators of Pollutant Emissions

With the environmental impact and human toxicity indicators, the potential risks generated by exhaust gases from power plants are evaluated. The global warming potentials (GWP), acid rain formation (ARP), smog formation (SFP) and human toxicity (HPT) are presented in Table 5 [31]. This indicates that CO has a global warming potential greater than CO_2_, that is, for every kilogram of CO there are 3 kg of CO_2_ equivalent; however, both have no potential for acid rain and smog formation. For each kilogram of unburned C*_n_*H*_m_*, there are 21 kg of CO_2_ equivalent and 0.015 kg of NO*_x_* equivalent, that is, for the potential for smog formation, the most important ozone-forming process in the atmosphere is the photo- NO*_x_* dissociation and is the reference compound to express the smog-forming potential of any smog-forming product or emission. Moreover, it is noted that only CO and NO_2_ have an effect on HPT. NO*_x_* has the highest potential in each type, where NO_2_ has a value at each potential and NO has a higher ARP. Table 6 presents the mathematical models to determine the environmental and human toxicity indicators.

## 3. Operating Conditions

The dead or reference state is established at *T*_0_ = 25 °C and *P*_0_ = 1.013 bar; and on the other hand, in Table 7, Table 8 and Table 9, the operating conditions of the CC with postcombustion are given [27].

Figure 5 shows that the highest amounts of heat transferred by the combustion gases are in the HPEV, and in the HPSH2 + IPRH2, with 23.35% and 19.02% of the total heat flux transferred in the HRSG, respectively. The hot approach temperature difference, *T_g_*_4_−*T_v_*_1_, is 91.77 °C. For the postcombustion section, there is no heat transfer to the water stream (liquid-vapor), the effect of postcombustion is manifested with an increase in the temperature of the exhaust gases from *g*5 to *g*6, at 104.43 °C caused by supplying the heat released by combustion in this section, of 86,371.15 kW.

In Figure 6, the hatched area between the flow gas and water streams represents the exergy losses due to heat transfer in the HRSG; HPEV and HPSH2 + IPRH2 have the highest exergy losses, with 26.26% and 22.77%, respectively. Moreover, the hatched area of the triangle *g15-0-a* is the exergy losses due to the waste of heat by the CC exhaust gases and represents 6.28% of exergy losses. In relation to the total heat supplied in the HRSG, for the postcombustion section, the heat of combustion represents 14.57%, and the IPSH and LPSH have the lowest heat transfers to the water stream (liquid-vapor) with 0.34% and 0.93%, respectively.

Table 10 presents the exergetic flow rates of the resource (*Ḟ*), the product (*Ṗ*), the irreversibility (*İ*) and the residue (*Ṙ*), as well as the exergetic efficiency (*η_ex_*) of the components of the CC with postcombustion. In the second column (*Ḟ*), we have that the highest exergy resource flow rate is in the CCH since it is the component where the greatest external resource of the CC enters, 15.36 kg/s of fuel, to generate the combustion gases at the temperature and pressure conditions required by the GT and the pumps present the lowest exergy flow rates of the resource, due to their supplied powers, of which the low-pressure pump has the lowest flow. The exergetic flow rate of the product (*Ṗ**)* is presented in the third column, where the turbine of the gas turbine system is the component that generates the highest exergetic flow rate of the product in the CC, with the power supplied to the compressor and the power output from the GT. The fourth column corresponds to the irreversibility flow rate (*İ*) where, the CCH is the component with the greatest contribution, due to the great loss of exergy that is had due to the need to reduce the temperature of the combustion gases, for their entry into the turbine of the GT system. The exergetic flow rate associated with the residue is presented in the fifth column. For this system, only those associated with the exhaust gases that come out of the chimney and the heat dissipated by the condenser towards the environment are studied. In the sixth column, the values of the exergy destruction factor are presented, this is an indicator that evaluates the exergy losses due to the irreversibility flow of equipment, in relation to the exergy flow of its resource (*İ/**Ḟ*), where the equipment is the smallest factor found in t, HPST and IPST, and therefore, they have the greatest use of their resource. Likewise, the seventh column shows the exergetic efficiency (*η_ex_*), where the gas system turbine has the highest and the lowest exergetic efficiencies are in the LPEC, the postcombustion section and the LPSH; The HPSH1 + IPRH1 section has the highest exergetic efficiency within the HRSG, due to its lower exergy factor destroyed by internal irreversibilities (*İ/**Ḟ*) of the heat exchange sections (combustion gases with liquid-vapor stream) in the HRSG, with 7.54%; M and m do not have exergetic efficiency because they represent stream junction points.

Figure 7 shows the Grassmann diagram of the CC and indicates that the external resources are the exergy flow rates of the fuel supplied in the CCH and in the HRSG, with 89.80% and 10.20%, respectively. The highest irreversibility flow rates are generated in the compressor, in the heat exchange process in the CCH and in the turbine with 19.99%; followed by the combustion process in the CCH, with 11.28%, being exergetic losses inherent in the combustion, related to the Carnot factor with the adiabatic flame and environmental temperatures. For the combustion process in the postcombustion section, there is a flow of inherent irreversibilities, with 1.25%. For the SC, the highest irreversibility flow rates are in the LPST and condenser, with 1.19% and 1.17%, respectively; and the lowest irreversibility flow rates correspond to the three pumps, with 0.0221%, since they convert ordered energy to a disordered form of energy, that is, the powers supplied to the pumps increase the exergy flow rates of the saturated liquid entering themselves, by increasing their pressure. There are two residuals or external irreversibilities, the exergy flow rates of the exhaust gases emitted into the environment and the heat flow dissipated by the condenser, with 0.84% and 0.90%, respectively. It also reveals that the exergetic efficiency of CC is 53.15%.

Table 11 presents the allocation factors for the cost of formation of the residue gas from the HRSG, *μ_k_*, and the heat rejected into the environment by the condenser, *β_k_*, in the components. With the criteria used, the components of the GT do not contribute to the formation of the heat dissipated by the condenser, so their allocation factor is *β_k_* = 0 and they do not have an allocation cost for this residue; on the other hand, the components of the SC do not intervene in the formation of the exhaust gases from the HRSG, and therefore their allocation factor is *μ_k_* = 0 and they do not have an allocation cost for this residue. Furthermore, for the exhaust gases, only the GT and HRSG components intervene in the formation of the cost of this residue; the CCH contributes with a factor *μ* of 0.6431 to the formation of its cost, followed by the HRSG (with all its sections) with a factor *μ* of 0.1481. In the HRSG, the postcombustion section has the highest contribution with a factor *μ* of 0.1418, because the allocation factors per component are directly proportional to its irreversibility; For example, CC has the highest allocation factor due to the formation of the cost of this residue, since it has the highest irreversibility flow rate. For the heat rejected to the environment, the GT equipment does not contribute to the formation of the cost of this residue, and the HRSG and LPST contribute more to the formation of this cost, with β factors of 0.7335 and 0.1521, respectively.

The total exergetic costs of the residues are higher than their exergetic flow rates, due to the irreversibilities accumulated during their formation process, as presented in Table 7 and Table 8, and when comparing these values there is an increase of 39.36% and 290.11%, respectively.

Considering that the cost of natural gas is *c_f_*_1_
*= c_f_*_2_
*=* 6.47 USD/GJ [19], the cost of air of *c_g_*_1_
*=* 0 and the cost of replacement water of *c_v_*_7_
*=* 0, the exergy and exergoeconomic costs are obtained, unit exergy and exergoeconomic costs for each stream, and the exergoeconomic operating costs of each component are determined.

Table 12 presents the exergoeconomic costs of the resource, product and residue attributed to the cost of the resource of the productive components of the CC, as well as the exergoeconomic unit costs and exergoeconomic operating costs of these components. The exergoeconomic cost values of the components are calculated with Equations (7)–(9), where the product cost of each component is equal to the cost of its resource and the proportion of its residue formation costs (exhaust gases and heat dissipated in the condenser). The second column shows that the turbine has the highest exergoeconomic cost of the resource with a value of 8061 USD/h, since this resource has a high exergy flow rate and a large amount of accumulated irreversibilities; followed by the CCH with an exergoeconomic cost of the resource of 6992 USD/h, associated with the exergy flow rate of the resource external to the CC. In the third column, the GT turbine has the highest exergoeconomic cost of the product with a value of 8.069 USD/h, mainly due to the exergoeconomic cost of its resource, since the exergoeconomic cost of its product is equal to the sum of the cost exergoeconomic of its resource and the exergoeconomic cost of allocation of the residue (exhaust gases), for intervening in the process of its formation. The CCH presents the highest exergoeconomic cost of the residue attributed to the formation of exhaust gases, 67.12 USD/h, since this component also has the highest irreversibility flow rate, and consequently, the highest allocation factor due to the formation of exhaust gases. The chimney has an exergoeconomic cost of its residue (exhaust gases) of 104.35 USD/h, which is the sum of the exergoeconomic costs of allocation of the productive components of the GT and HRSG (gas side) that intervene in the formation of this residue.

In column five, the HPEV has the highest exergoeconomic cost of the residue attributed to the formation of heat dissipated to the surroundings with a value of 53.78 USD/h, due to its higher allocation factor, due to the formation of this residue, in relation to the productive components of the HRSG (liquid-vapor side) and the SC; and the condenser has an exergoeconomic cost of its residue (heat dissipated to the surroundings) of 315.84 USD/h, which is the sum of the exergoeconomic costs of allocation of the productive components of the HRSG (liquid-vapor side) and of the SC that they intervene in the formation of this residue. Finally, the sixth and seventh columns present the unit exergoeconomic costs of the resource and of operation, respectively. The LPD and the pumps are the components that have the highest unit exergoeconomic costs of the resource with a value of 6.62 USD/h and 6.07 USD/h, respectively, which indicates that the process of obtaining their resources is the most inefficient; while, the CCH has an exergoeconomic operating cost of 1344 USD/h, which is the highest, mainly due to its higher irreversibility flow rate; and the lowest operating costs are found in LPP and IPP, due to their lower irreversibility flow rates.

## 4. Comparative Analysis of Cycle Combined with and without Postcombustion

Table 13 and Table 14 present thermodynamic, environmental, and human toxicity indicators of CC with and without postcombustion, where the indicators of CC without postcombustion are based on the paper by Lugo et al. [9]. The power output, the thermal and exergetic efficiencies, the mass flow rates of air and fuel in the GT are the same in both cycles because the operating conditions in this GT are the same. Likewise, the power output from the SC and the heat flow supplied by the combustion gases in the HRSG, the CC with postcombustion has the highest values with 171.67 MW (27.65% in relation to the CC without postcombustion) and 542.90 MW (42.41% with relation to CC without postcombustion), respectively.

The CC with postcombustion generates a higher power output of 310.87 MW, which represents an increase of 13.58%; the total mass flow rate of fuel in the cycle is higher in the combined cycle with postcombustion with 17.10 kg/s (11.32%), due to the mass flow rate of fuel injected in the postcombustion section, since the mass flow rate of fuel supplied in the GT is the same. The mass flow rates of steam in the SC are higher in the combined cycle with postcombustion, and therefore its power output in the SC increases in relation to the combined cycle without postcombustion; the thermal and exergetic efficiencies are higher in the CC without postcombustion with 54.30% and 62.15%, respectively, indicating a difference of 1.73% with the other combined cycle, due to the fact that their efficiencies in the SC are also lower, while, their GT efficiencies are equal.

The exergoeconomic cost of the power output in the GT is higher in the CC without postcombustion, however, the exergoeconomic costs of the power output in the SC and of residue formation (combustion gases at the outlet of the HRSG and dissipated heat by the condenser) are greater in the CC with postcombustion; the unit exergoeconomic cost of the power output in the GT is lower in the CC with postcombustion, indicating greater efficiency in the process of obtaining this energy stream, and the unit exergoeconomic costs of the power output in the SC and of the formation of the residues (combustion gases leaving the HRSG and heat dissipated by the condenser) are higher in the CC with postcombustion. The EOC of the CC with post-combustion is higher at 4374 USD/h, representing an increase of 54.53% in relation to the other cycle, mainly due to its higher irreversibility flow rate, despite the fact that the exergoeconomic operating costs of the GT are equal in both cycles combined. Furthermore, the exergoeconomic operating costs of the HRSG and SC are higher in the CC with postcombustion, since there are higher irreversibility flow rates. The environmental indicators of global warming, smog formation, acid rain formation and human toxicity are higher in the CC with postcombustion, due to its higher fuel consumption, despite the fact that more power is generated in the CC with postcombustion, that is, for each kWh produced, 373.25 g_CO_2_eq_, 2.53 g_NOxeq_, 2.62 g_SO_2_eq_ and 0.00067 g_Pbeq_ are generated in the CC without postcombustion, and the increments of these values in the CC with postcombustion are 19.52 g_CO_2_eq_, 0.14 g_NOxeq_, 0.14 g_SO_2_eq_ and 0.000034 g_Pbeq_ for each kWh produced.

## 5. Conclusions

The combined cycle with postcombustion has a power output ratio between the gas turbine and the steam cycle of 1.35/1, and the combined cycle without postcombustion has a power output ratio of 1.87:1, that is, the combined cycle with postcombustion increases the power output by 13.58%; however, its thermal and exergetic efficiencies decreased 0.94% and 1.08%, respectively, its environmental indicators increased (*I_GW_*-5.22%, *I_SF_*-5.53% and *I_AR_*-5.30%), its human toxicity indicator increased 5.07% and its exergoeconomic operating cost increased by 54.53%.

For the natural gas composition and the dead state temperature considered, the irreversibility flow rates in the combustion chamber represent 11.28% of the exergy losses of the combined cycle with postcombustion. These irreversibility flow rates are associated with the adiabatic flame temperature, which depends on the composition of the fuel gas, and on the dead state temperature (ambient temperature). The composition of the fuel gas is therefore a very important variable to determine the exergoeconomic, environmental and human toxicity indicators. Then the proper operation and maintenance of the fuel gas conditioning system and the combustion chamber are important to avoid increasing irreversibility flow rates in the combustion chamber. Since the combustion chamber presents the highest irreversibility flow rate, this component has the lowest exergetic efficiency and the highest exergoeconomic operating costs.

In this work, a criterion has been proposed to assign the cost of residue formation to the productive components related to its formation. This criterion is based on the irreversibilities generated in each productive component that participates in the formation of a residue. The criterion is an extension of the entropy change criterion because in addition to the entropy changes it also includes the exergy flow rates associated with the transfer of heat through the limits of the energy system. With these two criteria, all the productive components of each cycle take part in the formation of their respective residues. Then, the combined cycle generates two residues, one is the exhaust gases and the other the heat dissipated by the condenser, and with the criterion of irreversibilities generated to assign the cost of residue formation to the productive components involved in its formation, it is concluded that the cost of formation of the exhaust gases is formed in all the productive components of the gas turbine and the heat recovery steam generator, and the cost of formation of the heat dissipated by the condenser is formed in all the productive components of the cycle steam, and with postcombustion, the cost of formation of the exhaust gases increased 3.56% and the cost of formation of the heat dissipated by the condenser increased 8.38%.

For the exergoeconomic analysis in the combined cycle with postcombustion, the highest exergoeconomic costs of the resource and product are of the expansion turbine, of the gas turbine unit, because its resource has a high exergy flow and there is a large amount of irreversibilities accumulated during its obtaining process; the highest exergoeconomic cost of the residue attributed to the formation of exhaust gases is found in the combustion chamber, since it has the highest irreversibility flow, and therefore, it has the highest allocation factor due to the formation of combustion gases; the component with the highest exergoeconomic cost of the residue attributed to the formation of heat dissipated to the surroundings is the high pressure evaporator, due to its higher allocation factor, due to the formation of this residue, in relation to the components of the heat recovery steam generator (liquid-vapor side) and steam cycle; the exergoeconomic cost of the condenser residue is the sum of the exergoeconomic costs of allocation of the productive components of the heat recovery steam generator (liquid-vapor side) and of the steam cycle involved in the formation of this residue; the highest unit exergoeconomic costs of the resource are presented in the domes and pumps, indicating that the process of obtaining their resources are the most inefficient; the component with the highest exergoeconomic operating cost is the combustion chamber, mainly due to its greater irreversibility flow rate; and the lowest exergoeconomic operating costs are found in low pressure and intermediate pressure pumps, due to their lower irreversibility flow rates.

For the comparative analysis of combined cycles with and without postcombustion, the combined cycle without postcombustion has the highest exergoeconomic cost of the power output in the gas turbine, however, the combined cycle with postcombustion has the highest exergoeconomic costs of the power output in the cycle of steam and the formation of both residues; the unit exergoeconomic cost of the power output in the gas turbine is lower in the combined cycle with postcombustion, indicating greater efficiency in the process of obtaining this energy stream, and the unit exergoeconomic costs of the power output in the steam cycle and residue formation are higher in the combined cycle with postcombustion; the combined cycle with postcombustion has the highest exergoeconomic operating cost, mainly due to its higher irreversibility flow rate, despite the fact that the exergoeconomic operating costs of the gas turbine of both cycles is the same; the exergoeconomic operating costs of the heat recovery steam generator and steam cycle are higher in the combined cycle with postcombustion, since there are higher irreversibility flow rates; and the environmental indicators of global warming, smog formation, acid rain formation and human toxicity are higher in the combined cycle with postcombustion, due to its higher fuel consumption, despite the fact that more power output in the postcombustion cycle, these differences being 5.22%, 5.53%, 5.30% and 5.07%, respectively.

According to the indicators, it is concluded that a combined cycle with postcombustion can be used when there is a greater demand for power output, however, by having a higher fuel consumption, its environmental indicators are increased (global warming, formation of acid rain and smog), its total production costs (power output from the combined cycle) and its exergoeconomic operating costs.

## Figures and Tables

**Figure 1 entropy-24-00636-f001:**
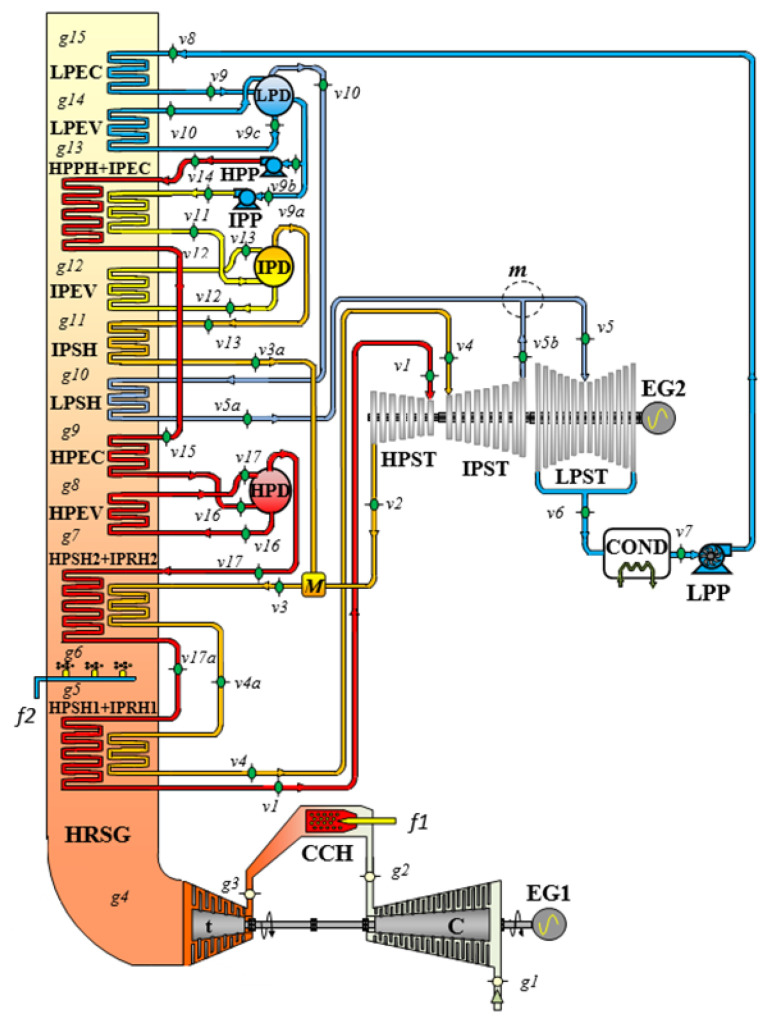
Combined Cycle with postcombustion.

**Figure 2 entropy-24-00636-f002:**
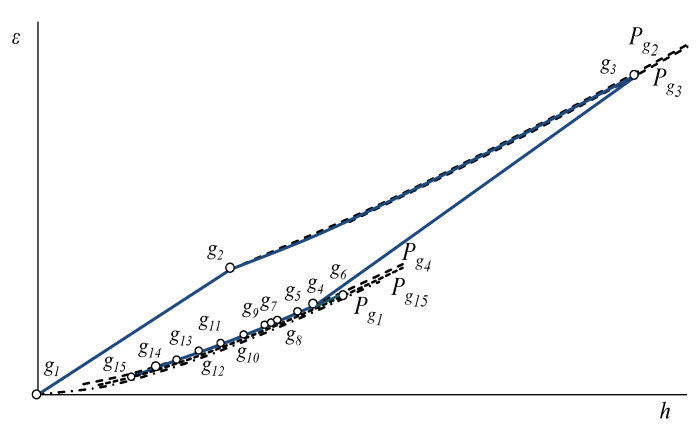
Exergy–enthalpy diagram of the gas turbine cycle and of the HRSG (gas side).

**Figure 3 entropy-24-00636-f003:**
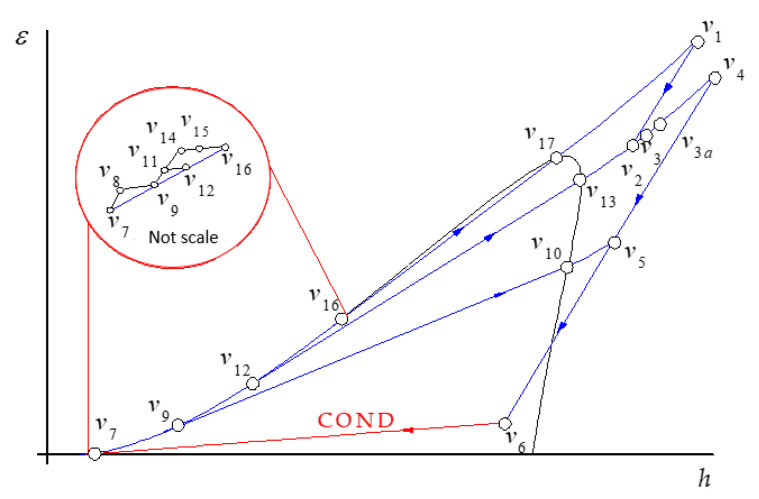
Exergy–enthalpy diagram steam cycle.

**Figure 5 entropy-24-00636-f005:**
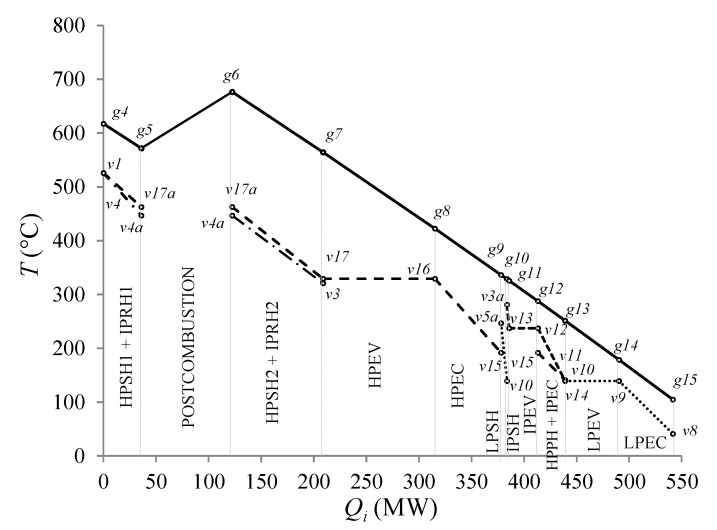
Temperature profile of the HRSG.

**Figure 6 entropy-24-00636-f006:**
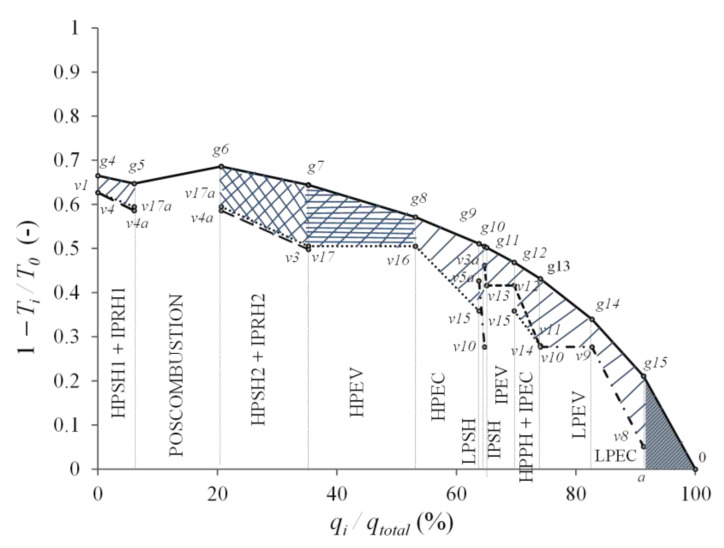
Carnot Coefficient profile of the HRSG.

**Figure 7 entropy-24-00636-f007:**
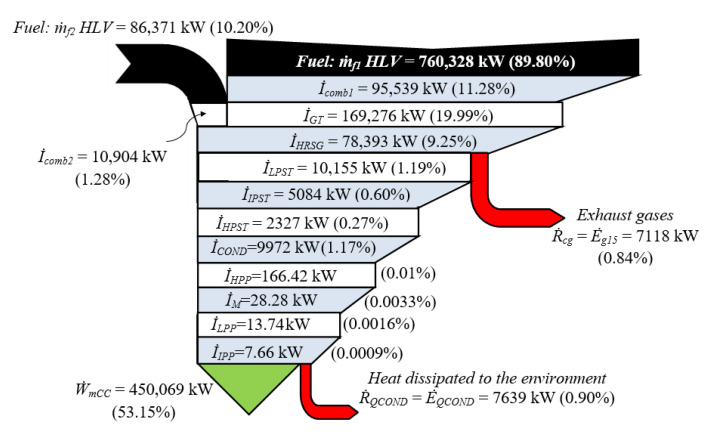
Grassmann diagram of combined cycle with postcombustion.

**Table 1 entropy-24-00636-t001:** Volumetric composition and mass fraction of the exhaust gases from the combined cycle with postcombustion.

Exhaust Gases	*δ_i_* (kmol*_i_*/kmol*_f_*)	*X_i_′* (%)	*f_i_′* (kg_*i*_/kg_gc_)
CO_2_	0.0472	4.70	0.07345
H_2_O	0.1025	10.22	0.06525
N_2_	0.7473	74.41	0.74011
O_2_	0.1063	10.59	0.12031
CO	0.000377	0.0375	0.00037
C*_n_*H*_m_*	0.000017	0.0001	0.00001
NO*_x_*	0.000428	0.0424	0.00046

**Table 2 entropy-24-00636-t002:** Exergoeconomic costs balance equations in the GT and the postcombustion section.

Component	Exergoeconomic Costs
External resources	$\Pi_{g1}^{}=c_{g1}^{}{\dot{E}}_{g1}^{}\;$ $,\;\Pi_{f1}^{}=c_{f}^{}{\dot{E}}_{f1}^{}\;$ $,\;\Pi_{f2}^{}=c_{f}^{}{\dot{E}}_{f2}^{}\;$
Compressor	$\Pi_{W_{C}}^{}+\mu_{C}\Pi_{g15}^{}+\beta_{C}\Pi_{Q_{COND}}^{}=\Pi_{g2}^{}-\Pi_{g1}^{}$ $\mu_{C}=\frac{{\dot{I}}_{C}}{{\dot{I}}_{cg}}$ $,\;\beta_{C}=0$
Combustion chamber	$\Pi_{f1}^{}+\mu_{cc}\Pi_{g15}^{}+\beta_{cc}\Pi_{Q_{COND}}^{}=\Pi_{g3}^{}-\Pi_{g2}^{}$ $\mu_{cc}=\frac{{\dot{I}}_{cc}}{{\dot{I}}_{cg}}$ $,\;\beta_{cc}=0$ $\Pi_{g3}^{}\;{\dot{E}}_{g4}^{}\;=\Pi_{g4}^{}\;{\dot{E}}_{g3}^{}\;$
Turbine	$\Pi_{g3}^{}-\Pi_{g4}^{}+\mu_{t}\Pi_{g15}^{}+\beta_{3}\;\Pi_{Q_{COND}}^{}=\Pi_{W_{mGT}}^{}+\Pi_{W_{C}}^{}$ $\mu_{t}=\frac{{\dot{I}}_{t}}{{\dot{I}}_{cg}}$ $,\;\beta_{t}=0$ $\Pi_{W_{C}}^{}\;{\dot{E}}_{W_{mGT}}^{}=\Pi_{W_{mGT}}^{}\;{\dot{E}}_{W_{C}}^{}\;$
Poscombustion	$\Pi_{f2}^{}+\mu_{Poscomb}\Pi_{g15}^{}+\beta_{Poscomb}\Pi_{Q_{COND}}^{}=\Pi_{g6}^{}-\Pi_{g5}^{}$ $\mu_{Poscomb}=\frac{{\dot{I}}_{Poscomb}}{{\dot{I}}_{cg}}$ $,\;\beta_{Poscomb}=0$

**Table 3 entropy-24-00636-t003:** Exergoeconomic costs balance equations in the HRSG.

Component	Exergoeconomic Costs
HPSH1 + IPRH1	$\Pi_{g4}^{}-\Pi_{g5}^{}+\mu_{HPSH1+IPRH1}\Pi_{g15}^{}+\beta_{HPSH1+IPRH1}\;\Pi_{Q_{COND}}^{}=\Pi_{v1}^{}-\Pi_{v17a}^{}+\Pi_{v4}^{}-\Pi_{v4a}$ $\mu_{HPSH1+IPRH1}=\frac{{\dot{F}}_{HRSG_{HPSH1+IPRH1}}-{\dot{E}}_{{\dot{Q}}_{HPSH1+IPRH1}}}{{\dot{I}}_{cg}}$$,\;\beta_{HPSH1+IPRH1}=\frac{{\dot{E}}_{{\dot{Q}}_{HPSH1+IPRH1}}-{\dot{P}}_{HRSG_{HPSH1+IPRH1}}}{{\dot{I}}_{v}}$, $\Pi_{g3}^{}{\dot{E}}_{g5}=\Pi_{g5}^{}{\dot{E}}_{g3}$
HPSH2 + IPRH2	$\Pi_{g6}^{}-\Pi_{g7}^{}+\mu_{HPSH2+IPRH2}\Pi_{g15}^{}+\beta_{HPSH2+IPRH2}\;\Pi_{Q_{COND}}^{}=\Pi_{v17a}^{}-\Pi_{v17}^{}+\Pi_{v4a}^{}-\Pi_{v3}^{}$ $\mu_{HPSH2+IPRH2}=\frac{{\dot{F}}_{HRSG_{HPSH2+IPRH2}}-{\dot{E}}_{{\dot{Q}}_{HPSH2+IPRH2}}}{{\dot{I}}_{cg}}$ $,\;\beta_{HPSH2+IPRH2}=\frac{{\dot{E}}_{{\dot{Q}}_{HPSH2+IPRH2}}-{\dot{P}}_{HRSG_{HPSH2+IPRH2}}}{{\dot{I}}_{v}}$ $\Pi_{g6}^{}{\dot{E}}_{g7}=\Pi_{g7}^{}{\dot{E}}_{g6}$
HPEV	$\Pi_{g7}^{}-\Pi_{g8}^{}+\mu_{HPEV}\Pi_{g15}^{}+\beta_{HPEV}\;\Pi_{Q_{COND}}^{}=\Pi_{v17}^{}-\Pi_{v16}^{}$ $\mu_{HPEV}=\frac{{\dot{F}}_{HRSG_{HPEV}}-{\dot{E}}_{{\dot{Q}}_{HPEV}}}{{\dot{I}}_{cg}}$$,\;\beta_{HPEV}=\frac{{\dot{E}}_{{\dot{Q}}_{HPEV}}-{\dot{P}}_{HRSG_{HPEV}}}{{\dot{I}}_{v}}$, $\Pi_{g6}^{}{\dot{E}}_{g8}=\Pi_{g8}^{}{\dot{E}}_{g6}$
HPEC	$\Pi_{g8}^{}-\Pi_{g9}^{}+\mu_{HPEC}\Pi_{g15}^{}+\beta_{HPEC}\;\Pi_{Q_{COND}}^{}=\Pi_{v16}^{}-\Pi_{v15}^{}$ $\mu_{HPEC}=\frac{{\dot{F}}_{HRSG_{HPEC}}-{\dot{E}}_{{\dot{Q}}_{HPEC}}}{{\dot{I}}_{cg}}$$,\;\beta_{HPEC}=\frac{{\dot{E}}_{{\dot{Q}}_{HPEC}}-{\dot{P}}_{HRSG_{HPEC}}}{{\dot{I}}_{v}}$, $\Pi_{g6}^{}{\dot{E}}_{g9}=\Pi_{g9}^{}{\dot{E}}_{g6}$
LPSH	$\Pi_{g9}^{}-\Pi_{g10}^{}+\mu_{LPSH}\Pi_{g15}^{}+\beta_{LPSH}\;\Pi_{Q_{COND}}^{}=\Pi_{v5a}^{}-\Pi_{v10}^{}$ $\mu_{LPSH}=\frac{{\dot{F}}_{HRSG_{LPSH}}-{\dot{E}}_{{\dot{Q}}_{LPSH}}}{{\dot{I}}_{cg}}$$,\;\beta_{LPSH}=\frac{{\dot{E}}_{{\dot{Q}}_{LPSH}}-{\dot{P}}_{HRSG_{LPSH}}}{{\dot{I}}_{v}}$, $\Pi_{g6}^{}{\dot{E}}_{g10}=\Pi_{g10}^{}{\dot{E}}_{g6}$

**Table 4 entropy-24-00636-t004:** Exergoeconomic costs balance equations in the SC.

Component	Exergoeconomic Costs
IPSH	$\Pi_{g10}^{}-\Pi_{g11}^{}+\mu_{IPSH}\Pi_{g15}^{}+\beta_{IPSH}\Pi_{Q_{COND}}^{}=\Pi_{v3a}^{}-\Pi_{v13}^{}$ $\mu_{IPSH}=\frac{{\dot{F}}_{HRSG_{IPSH}}-{\dot{E}}_{{\dot{Q}}_{IPSH}}}{{\dot{I}}_{cg}}$ $,\;\beta_{IPSH}=\frac{{\dot{E}}_{{\dot{Q}}_{IPSH}}-{\dot{P}}_{HRSG_{IPSH}}}{{\dot{I}}_{v}},$ $\Pi_{g6}^{}{\dot{E}}_{g11}=\Pi_{g11}^{}{\dot{E}}_{g6}$
IPEV	$\Pi_{g11}^{}-\Pi_{g12}^{}+\mu_{IPEV}\Pi_{g15}^{}+\beta_{IPEV}\;\Pi_{Q_{COND}}^{}=\Pi_{v13}^{}-\Pi_{v12}^{}$ $\mu_{IPEV}=\frac{{\dot{F}}_{HRSG_{IPEV}}-{\dot{E}}_{{\dot{Q}}_{IPEV}}}{{\dot{I}}_{cg}}$ $,\;\beta_{IPEV}=\frac{{\dot{E}}_{{\dot{Q}}_{IPEV}}-{\dot{P}}_{HRSG_{IPEV}}}{{\dot{I}}_{v}},$ $\Pi_{g6}^{}{\dot{E}}_{g12}=\Pi_{g12}^{}{\dot{E}}_{g6}$
HPPH + IPEC	$\Pi_{g12}^{}-\Pi_{g13}^{}+\mu_{HPPH+IPEC}\Pi_{g15}^{}+\beta_{HPPH+IPEC}\;\Pi_{Q_{COND}}^{}=\Pi_{v12}^{}-\Pi_{v11}^{}+\Pi_{v15}^{}-\Pi_{v14}^{}$ $\mu_{12}=\frac{{\dot{F}}_{CRC_{\mathrm{Pr}ecAP+ECPI}}-{\dot{E}}_{{\dot{Q}}_{\mathrm{Pr}ecAP+ECPI}}}{{\dot{I}}_{cg}}$ $,\;\beta_{12}=\frac{{\dot{E}}_{{\dot{Q}}_{\mathrm{Pr}ecAP+ECPI}}-{\dot{P}}_{CRC_{\mathrm{Pr}ecAP+ECPI}}}{{\dot{I}}_{v}},$ $\Pi_{g6}^{}{\dot{E}}_{g13}=\Pi_{g13}^{}{\dot{E}}_{g6}$
LPEV	$\Pi_{g11}^{}-\Pi_{g12}^{}+\mu_{LPEV}\Pi_{g13}^{}+\beta_{LPEV}\;\Pi_{Q_{COND}}^{}=\Pi_{v10}^{}-\Pi_{v9c}^{}$ $\mu_{LPEV}=\frac{{\dot{F}}_{HRSG_{LPEV}}-{\dot{E}}_{{\dot{Q}}_{LPEV}}}{{\dot{I}}_{cg}}$$,\;\beta_{\mathrm{LPEV}}=\frac{{\dot{E}}_{{\dot{Q}}_{LPEV}}-{\dot{P}}_{HRSG_{LPEV}}}{{\dot{I}}_{v}}$, $\Pi_{g6}^{}{\dot{E}}_{g14}=\Pi_{g14}^{}{\dot{E}}_{g6}$
LPEC	$\Pi_{g14}^{}-\Pi_{g15}^{}+\mu_{LPEC}\Pi_{g15}^{}+\beta_{\mathrm{LPEC}}\;\Pi_{Q_{COND}}^{}=\Pi_{v9}^{}-\Pi_{v8}^{}$ $\mu_{LPEC}=\frac{{\dot{F}}_{HRSG_{LPEC}}-{\dot{E}}_{{\dot{Q}}_{LPEC}}}{{\dot{I}}_{cg}}$$,\;\beta_{\mathrm{LPEC}}=\frac{{\dot{E}}_{{\dot{Q}}_{LPEC}}-{\dot{P}}_{HRSG_{LPEC}}}{{\dot{I}}_{v}}$, $\Pi_{g6}^{}{\dot{E}}_{g15}=\Pi_{g15}^{}{\dot{E}}_{g6}$

**Table 5 entropy-24-00636-t005:** Potentials for contamination and human toxicity.

Emitted Gas	GWP (kg_CO_2_eq_/kg*_i_*)	ARP (kg_SO_2_eq_/kg*_i_*)	SFP (kg_NOxeq_/kg*_i_*)	HTP (kg_Pbeq_/kg*_i_*)
CO	3	0	0	0.00014
CO_2_	1	0	0	0
C*_n_*H*_m_*	CH_4_-21	0	0.015-CH_4_	0
NO*_x_*	NO_2_-40	1.07-NO 0.7-NO_2_	1	0.002-NO_2_

**Table 6 entropy-24-00636-t006:** Indicators of environmental impact and human toxicity.

Global warming indicator $I_{GWP}=3.6\times 10^{6}\frac{\left({\dot{m}}_{cg}+{\dot{m}}_{f2}\right)\sum_{i}{(f}_{i})\;({GWP}_{i})}{{\dot{m}}_{a}\left[(1+far)\;\frac{c_{Pcg}}{c_{Pa}}\;y\;\eta_{sit}\;\left(1-\frac{1}{\pi_{t}{}^{x_{cg}}}\right)-\frac{1}{\eta_{sic}}\;\left(\pi_{c}{}^{x_{a}}-1\right)\right]{+\dot{W}}_{mSC}\;}$
Smog formation indicator $I_{SFP}=3.6\times 10^{6}\frac{\left({\dot{m}}_{cg}+{\dot{m}}_{f2}\right)\sum_{i}{(f}_{i})\;({SFP}_{i})}{{\dot{m}}_{a}\left[(1+far)\;\frac{c_{Pcg}}{c_{Pa}}\;y\;\eta_{sit}\;\left(1-\frac{1}{\pi_{t}{}^{x_{cg}}}\right)-\frac{1}{\eta_{sic}}\;\left(\pi_{c}{}^{x_{a}}-1\right)\right]{+\dot{W}}_{mSC}\;}$
Acid rain indicator $I_{ARP}=3.6\times 10^{6}\frac{\left({\dot{m}}_{cg}+{\dot{m}}_{f2}\right)\sum_{i}{(f}_{i})\;({ARP}_{i})}{{\dot{m}}_{a}\left[(1+far)\;\frac{c_{Pcg}}{c_{Pa}}\;y\;\eta_{sit}\;\left(1-\frac{1}{\pi_{t}{}^{x_{cg}}}\right)-\frac{1}{\eta_{sic}}\;\left(\pi_{c}{}^{x_{a}}-1\right)\right]{+\dot{W}}_{mSC}\;}$
Human toxicity indicator $I_{HTP}=3.6\times 10^{6}\frac{\left({\dot{m}}_{cg}+{\dot{m}}_{f2}\right)\sum_{i}{(f}_{i})\;({HTP}_{i})}{{\dot{m}}_{a}\left[(1+far)\;\frac{c_{Pcg}}{c_{Pa}}\;y\;\eta_{sit}\;\left(1-\frac{1}{\pi_{t}{}^{x_{cg}}}\right)-\frac{1}{\eta_{sic}}\;\left(\pi_{c}{}^{x_{a}}-1\right)\right]{+\dot{W}}_{mSC}\;}$

**Table 7 entropy-24-00636-t007:** Ambient conditions and parameters for the operation of the GT.

*Ẇ_mGT_*, (MW)	139.2
*TIT or T_g_*_3_, (°C)	1300
*π_C_*, (-)	16
*η_sic_*, (-)	0.88
*η_sit_*, (-)	0.9
Δ*P_CCH_*/*P_g_*_2_, (%)	2
Δ*P_t_*/*P_atm_*, (%)	1
*T_amb_* and *T*_0_, (°C)	25
*P_atm_* and *P*_0_, (bar)	1.013

**Table 8 entropy-24-00636-t008:** Design parameters for the operation of the SC.

*T_v_*_1_, (°C)	525.8
*HP*, (bar)	127.38
*IP*, (bar)	32.06
*LP*, (bar)	3.53
*P_COND_*, (bar)	0.078
*η_sitv_*, (-)	0.88
*η_sip_*, (-)	0.85

**Table 9 entropy-24-00636-t009:** Pinch-point temperature differences in the HRSG.

Δ*T_ppLP_*, (°C)	39.43
Δ*T_ppIP_*, (°C)	50.24
Δ*T_ppHP_*, (°C)	93.42

**Table 10 entropy-24-00636-t010:** Resource, product, irreversibility and residue flow rates and exergetic efficiencies of CC components.

Components	*Ḟ* (kW)	*Ṗ* (kW)	*i* (kW)	*Ṙ* (kW)	*f_exd_* (-)	*η_ex_* (%)
Gas turbine (GT)						
C	279,167	253,544	25,623	0	0.0917	90.82
CCH	664,788	536,977	127,811	0	0.1922	80.77
t	573,410	557,567	15,842	0	0.0276	97.23
Heat recovery steam generator (HRSG)						
HPSH1 + IPRH1	24,403	22,563	1840	0	0.0754	92.45
Postcombustion	75,466.40	47,274.09	28,192.31	0	0.3735	62.64
HPSH2 + IPRH2	58,583	47,863	10,720	0	0.1829	81.70
HPEV	67,579	55,926	11,652	0	0.1724	82.75
HPEC	36,663	29,912	6750	0	0.1841	81.58
LPSH	3097.76	2144.84	952.92	0	0.3076	69.23
IPSH	1169.25	983.97	185.28	0	0.1584	84.15
IPEV	14,520	12,407	2113	0	0.1455	85.44
HPPH + IPEC	12,929	9397	3531	0	0.2731	72.68
LPEV	22,141	15,769	6371	0	0.2877	71.22
LPEC	16,179	10,096	6082	0	0.3759	62.40
Chimney	7118	0	0	7118	-	-
Steam cycle (SC)						
HPST	36,208	33,880	2327	0	0.0642	93.57
IPST	67,765	62,680	5084	0	0.0750	92.49
LPST	88,631	78,475	10,155	0	0.1145	88.54
M	104,732	104,732	0	0	-	-
LPP	54.43	40.69	13.74	0	0.2524	74.74
IPP	61.03	53.36	7.66	0	0.1255	87.43
LPP	1512.18	1345.76	166.42	0	0.1100	88.99
LPD	28,104	28,104	0	0	-	-
IPD	20,175	20,175	0	0	-	-
HPD	146,376	146,376	0	0	-	-
m	17,275.43	17,247.14	28.29	0	-	-
COND	15,871	-	8232	7639	0.5186	-

**Table 11 entropy-24-00636-t011:** Allocation factors of the costs of formation of residue of exhaust gases and heat rejected to the environment and exergetic costs of the residues.

Components	*µ* (-)	$R_{cg}^{*}(\mathrm{kW})$	β (-)	$R_{Q_{COND}}^{*}(\mathrm{kW})$
Gas turbine (GT)				
C	0.1289	1279	0	0
CCH	0.6431	6380	0	0
t	0.0797	790.94	0	0
Heat recovery steam generator (HRSG)				
HPSH1 + IPRH1	0.0004	4.01	0.0263	785.81
Postcombustion	0.1418	1407	0	0
HPSH2 + IPRH2	0.0005	5.13	0.1590	4741
HPEV	0.0014	14.30	0.1702	5075
HPEC	0.0006	6.48	0.0992	2956
LPSH	0.0002	1.98	0.0136	407.74
IPSH	0.0001	1.98	0.0021	64.98
IPEV	0.0002	2.53	0.0309	920.88
HPPH + IPEC	0.0005	4.22	0.0516	1539
LPEV	0.0008	8.46	0.0929	2769
LPEC	0.0013	13.03	0.0872	2599
Steam cycle (SC)				
HPST	0	0	0.0348	1039
IPST	0	0	0.0761	2270
LPST	0	0	0.1521	4534
LPP	0	0	0.0002	6.13
IPP	0	0	0.0001	3.42
HPP	0	0	0.0024	74.31
M	0	0	0.0004	12.63
Total	1.0	9920	1.0	29,801

**Table 12 entropy-24-00636-t012:** Exergoeconomic costs of resource, product and residue, and unit exergoeconomic and exergoeconomic operating costs.

Components	*Π_F_* (USD/h)	*Π_P_* (USD/h)	*Π_Rcg_* (USD/h)	*Π_RQCOND_* (USD/h)	*c_F_* (USD/GJ)	*COE* (USD/h)
Gas turbine (GT)						
C	4040.48	4053.94	13.45	-	4.02	370.85
CCH	6992.84	7059.96	67.12	-	2.92	1344.43
t	8061.54	8069.86	8.31	-	3.90	222.73
Heat recovery steam generator (HRSG)						
HPSH1 + IPRH1	343.09	351.46	0.04	8.32	3.90	25.87
Postcombustion	793.82	47,274.09	14.80	-	2.92	296.55
HPSH2 + IPRH2	858.78	909.08	0.05	50.24	4.07	157.15
HPEV	990.64	1044.58	0.15	53.78	4.07	170.81
HPEC	537.44	568.84	0.06	31.33	4.07	98.95
LPSH	45.41	49.75	0.02	4.32	4.07	13.96
IPSH	17.14	17.84	0.02	0.68	4.07	2.71
IPEV	212.85	222.64	0.02	9.75	4.07	30.97
HPPH + IPEC	189.53	205.89	0.04	16.31	4.07	51.77
LPEV	324.56	354.00	0.08	29.35	4.07	93.40
LPEC	237.17	264.85	0.13	27.54	4.07	89.16
Chimney	104.35	-	104.35	-	-	-
Steam cycle (SC)						
HPST	693.35	704.36	-	11.01	5.31	44.57
IPST	1277.89	1301.96	-	24.06	5.23	95.89
LPST	1738.52	1786.58	-	48.05	5.44	199.19
M	2054.36	2054.36	-	-	5.44	-
LPP	1.19	1.25	-	0.06	6.07	0.30
IPP	1.33	1.37	-	0.03	6.07	0.16
HPP	33.09	33.88	-	0.78	6.07	3.64
LPD	670.63	670.63	-	-	6.62	-
IPD	390.01	390.01	-	-	5.36	-
HPD	2928.38	2928.38	-	-	5.55	-
m	324.17	324.31	-	0.13	5.21	-
COND	315.84	-	-	315.84	-	-

**Table 13 entropy-24-00636-t013:** Thermodynamic, environmental, and human toxicity indicators of CC with and without postcombustion.

Indicators	CC without Postcombustion	CC with Postcombustion
Operating parameters of the CC		
*Ẇ_mGT_*, (MW)	278.4	278.4
*Ẇ_mSC_*, (MW)	134.48	171.67
*Ẇ_mCC_*, (MW)	273.68	310.87
*ṁ_a_*, (kg*_a_*/s)	626.81	626.81
*ṁ_f_*, (kg*_f_*/s)	15.36	17.10
*ṁ_HP_*, (kg*_v_*/s)	76.77	96.44
*ṁ**_IP_*, (kg*_v_*/s)	11.28	16.77
*ṁ**_LP_*, (kg*_v_*/s)	21.99	26.52
${\dot{Q}}_{HRSG}$, (MW)	381.22	542.90
Thermodynamic indicators		
*η_thGT_*, (%)	35.16	35.16
*η_exGT_*, (%)	40.25	40.25
*η_thSC_*_,_ (%)	35.27	31.94
*η_exSC_*, (%)	65.15	52.11
*η_thCC_*_,_ (%)	54.30	53.36
*η_exCC_*_,_ (%)	62.15	61.07

**Table 14 entropy-24-00636-t014:** Thermodynamic, environmental, and human toxicity indicators of CC with and without postcombustion.

Indicators	CC without Postcombustion	CC with Postcombustion
Environmental and human toxicity indicators		
*I_GW_*, (g_CO_2_eq/_kWh)	373.25	392.77
*I_SF_*, (g_NOxeq_/kWh)	2.53	2.67
*I_AR_*, (g_SO_2_eq_/kWh)	2.64	2.78
*I_HT_*, (g_Pbeq_/kWh)	0.00067	0.000704
Exergoeconomic indicators		
Π*_WmGT_*, (USD/h)	4058	4029
*c_WmGT_*, (USD/GJ)	4.05	4.02
Π*_WmST_*, (USD/h)	2933	3757
*c_WmST_*, (USD/GJ)	6.05	6.07
Π*_cg_*, (USD/h)	98.95	104.35
*c_cg_*, (USD/GJ)	3.93	4.07
Π*_QCOND_*, (USD/h)	291.40	315.84
*c_QCOND_*, (USD/GJ)	6.47	11.48
*EOC_GT_*, (USD/h)	1 941	1941
*EOC_HRSG_*,(USD/h)	618.25	1402
*EOC_SC_*, (USD/h)	271.12	1031
*EOC_CC_*, (USD/h)	2830.37	4374

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
