# Peer review of "Comparison of the Parameters of the Exergoeconomic Environmental Analysis of Two Combined Cycles of Three Pressure Levels with and without Postcombustion"

_entropy, 2022, doi:10.3390/e24050636_

Round 1
Reviewer 1 Report
I attach a pdf file with the comments.

Author Response
"Please see the attachment."

Reviewer 2 Report
The paper is well written and the scope of the study corresponds to pure thermodynamic and economic aspects together with environmental. However, I have a few suggestions before making any decision.
- The introduction needs more details about motivation, literature survey, and novelty of the proposed work.
- Details of Figure 1, 2, and 3 are not enough. Please rewrite your system's description so that readers understand the situation and proposed methodology.
- More details of economic evaluation are required.
- Life cycle assessment is suggested to claim environmental benefits
Author Response
"Please see the attachment."
